# UX in AR-Supported Industrial Human–Robot Collaborative Tasks: A Systematic Review

Riccardo Karim Khamaisi [1], Elisa Prati [1], Margherita Peruzzini [1,*], Roberto Raffaeli [2] and Marcello Pellicciari [2]

1 Department of Engineering "Enzo Ferrari", University of Modena and Reggio Emilia, 41125 Modena, Italy; rcrkhamaisi@unimore.it (R.K.K.); elisa.prati@unimore.it (E.P.)
2 Department of Sciences and Methods for Engineering, University of Modena and Reggio Emilia, 42122 Reggio Emilia, Italy; roberto.raffaeli@unimore.it (R.R.); marcello.pellicciari@unimore.it (M.P.)
* Correspondence: margherita.peruzzini@unimore.it

**Abstract:** The fourth industrial revolution is promoting the Operator 4.0 paradigm, originating from a renovated attention towards human factors, growingly involved in the design of modern, human-centered processes. New technologies, such as augmented reality or collaborative robotics are thus increasingly studied and progressively applied to solve the modern operators' needs. Human-centered design approaches can help to identify user's needs and functional requirements, solving usability issues, or reducing cognitive or physical stress. The paper reviews the recent literature on augmented reality-supported collaborative robotics from a human-centered perspective. To this end, the study analyzed 21 papers selected after a quality assessment procedure and remarks the poor adoption of user-centered approaches and methodologies to drive the development of human-centered augmented reality applications to promote an efficient collaboration between humans and robots. To remedy this deficiency, the paper ultimately proposes a structured framework driven by User eXperience approaches to design augmented reality interfaces by encompassing previous research works. Future developments are discussed, stimulating fruitful reflections and a decisive standardization process.

**Keywords:** User eXperience; human–robot interaction; human–robot collaboration; human-centered design; augmented reality; human factors





## 1. Introduction

The creation of intelligent, assisted, and automated machines is characterizing the modern factory aiming at two main aspects: a more conscious distribution of roles between machines and humans, and a more flexible process control to achieve an efficient and optimized production. In this context, high standards of quality, production flexibility, and innovation push towards human-centered design (HCD) approaches, focused on the centrality of the human factors (HF). HF refers to environmental, organizational, and job-related aspects, as well as human individual characteristics, which can highly affect health and safety during the interaction with current technologies. Introducing HF in the design process is the scope of HCD, which is defined as "*an approach to systems design and development that aims to make interactive systems more usable by focusing on the use of the system and applying human factors/ergonomics and usability knowledge and techniques*" [1]. Today, HCD can be generically used for any type of applications to guarantee the satisfaction of user needs and the coherence with the ergonomics principles while designing any type of human–system interaction. HCD enables new ways to define requirements and recommendations to properly design complex systems according to a user-oriented approach. The final goal is to guarantee a valuable User eXperience (UX), which involves "*the user's perceptions and responses that result from the use and/or anticipated use of a system product or service*" [1], including usability in terms of "*the achievement of specified goals with effectiveness,*

*efficiency and satisfaction in a specified context of use*", but also considering users' emotions and affections [2].

The current frameworks related to the application of HF and HCD in system design need to be further developed with the advent of Operator 4.0 (O4.0) concept, framing a smart and skilled operator performing highly specialized tasks aided by emerging technologies as and if needed [3], in order to reshape the industrial tasks based on the human-machine partnership and to renovate the industrial systems according to Industry 4.0 paradigm. Indeed, the O4.0 idea is introducing new assistive technologies, such as augmented reality (AR), virtual reality (VR), or mixed reality (MR) in modern industries, making them enabling technologies for the design and development of an effective human–machine cooperation. However, to achieve such challenging objectives, technologies must be centered on the figure of the modern Operator 4.0 according to new framework, able to focus on the interface design for collaborative tasks, involving humans and robots. Primarily, a precise distinction among such technologies can be summarized as follows:

- Augmented reality, as defined by Azuma et al., "supplements the real world with virtual (computer-generated) objects that appear to coexist in the same space as the real world" [4];
- Virtual reality implies a full immersion into a fictious and digitally generated world which shuts out completely the physical world [5];
- Mixed reality combines both the previous technologies while enabling a strict interaction between the digital and physical world. Thus, the user interaction with the computer-generated environment provides feedbacks and vice versa [6].

Secondly, attention has to be paid to the technological development of modern companies, where novel forms of support and training can be introduced to enrich the operator's knowledge and encouraging the proper use of new, emerging tools, such as robots [7]. Considering all that, the proper design of AR and VR interfaces becomes crucial to promote the new-born paradigm of the Operator 4.0. In order to achieve higher task precision and market responsiveness, industrial collaborative robots and AR devices are gradually entering the shop floor level to assist operators [8]. Contextually, designing a collaborative working environment for O4.0 requires the adoption of the HCD approach in order to consider the O4.0 know-how and know-to-cooperate: the former refers to human capability to run the process, whilst the latter deals with their attitude to cooperate with other agents [9]. Hence, agents' intentions, action's adaptability, and safety concerns are steadily part of human–robot interaction (HRI). The latter is a field of study concentrated on the design of robotic systems for use by or with humans which seeks to improve the human–machine collaboration while developing innovative and usable user interfaces. Finally, the analysis of the state of the art highlights the need for defining a new HCD framework tackling the new O4.0 requirements to improve the design of AR interfaces, according to UX interface design, but applied them specifically for HRI scopes. Therefore, this review also proposes a framework to design AR interfaces as a natural outcome of this review work, due to the lack in the existing literature.

The review moves from the analysis of the different levels of HRI, namely coexistence, cooperation, and collaboration [10]. Coexistence refers to humans and robots sharing common workspace and time, but using different resources. Cooperation is characterized by a common workspace, time, and shared aim, with sequential or simultaneous tasks, on the same resources, but does not involve a direct contact between humans and robots. Finally, collaboration is the highest level of interaction that envisages common workspace, time, and shared aim, with sequential or simultaneous tasks on the same resources, involving a direct physical contact between humans and robots.

This distinction demonstrates how specific UX issues can be identified for each level of HRI. Thus, as shown in Figure 1 and according to the distinction just made, technologies such as AR could be selectively used to support specific targeted tasks of the human–robot interaction. Moreover, regardless of HRI nature, in [10] the authors suggest to integrate a human-centered view to the robot-centered and robot cognition-centered views, meaning

to harmonize the HF and human–machine interaction principles with technological and decisional capability aspects. Based on [11], it could be stated that:

- The human-centered view is primarily concerned "*with how a robot can fulfil its task specification in a manner that is acceptable and comfortable to humans*";
- The robot-centered view "*emphasizes the view of a robot as a creature, i.e., an autonomous entity that is pursuing its own goals based on its motivations, drives and emotions, whereby interaction with people serves to fulfil some of its 'needs'*";
- The robot-cognition view considers "*the robot as an intelligent system (in a traditional AI sense), i.e., a machine that makes decisions on its own and solves problems it faces as part of the tasks it needs to perform in a particular application domain.*".

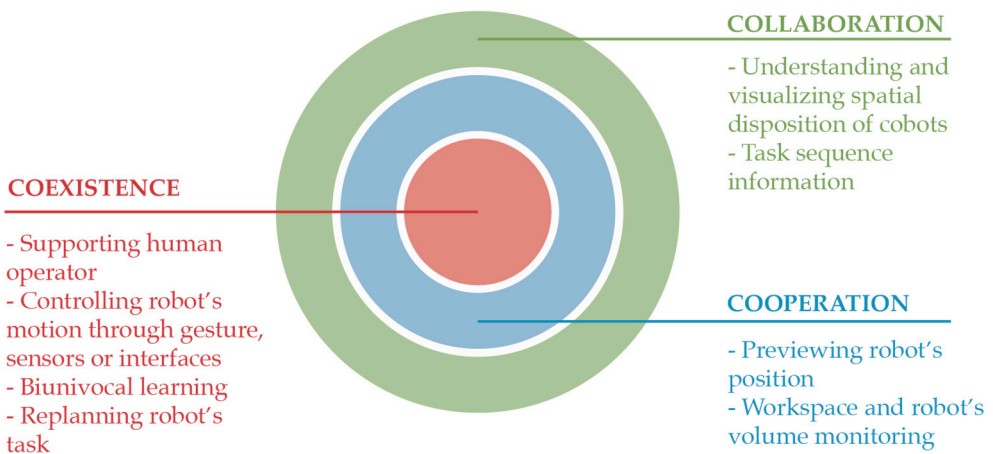

**Figure 1.** Main AR applications areas according to the three levels of HRI.

Such considerations are valid even if multiple humans and robots are involved in the interaction. All these terminological and conceptual distinctions demonstrate the intrinsic complexity of a HRI task and the need of a structured approach to appropriately encompass all its peculiarities.

However, in practice, collaborative robots (i.e., "*cobots*") are currently regulated by the ISO 10218 technical specification document [12], providing a precise interpretation of their roles and natures, and defining the safety requirements for industrial collaborative systems. The main focus is actually for safety as addressed by ISO 15066 [13], neglecting other human implications. We could say that the focus is on the robot-centered view, and only marginally on the robot cognition-centered view where the human-centered aspects are only considered for safety purposes.

In this context, the use of AR technologies as task support tools force us to pay attention to the human-centered view by supporting HRI at different levels (i.e., coexistence, cooperation, and collaboration), thanks to the creation of specific, contextual human knowledge on the ongoing process, and the promotion of know-how development and know-to-cooperate abilities [9]. Indeed, an effective AR interface should define their contents according to the level of interaction realized between the operator and the robots, since each level is characterized by a different form of interaction that requires specific features in the AR interface to properly support the tasks. Indeed, AR introduces digitized information into the real working environment to augment the UX, promoting system usability and object visibility, while reducing the operator's physical and cognitive workload. Current AR applications are provided mainly through tablets since head-mounted display are still far from being industrially reliable, especially as for ergonomic aspects. The state of art in literature regarding AR-supported HRI highlighted different application areas: visualizing robot actions or faults to support troubleshooting [14]; controlling robots combining head and eye gaze; visual simultaneous localization and mapping algorithms [15]; understanding the impact of AR cues on human attention [16]; supporting human–robot

collaborative assembly [17]; providing workspace and robot's volume monitoring [8,18]; improving interaction efficiency by reducing the physical strength (especially in heavy-duty industries) or letting older people to continue working in production facilities [19]; and by helping operators to have an immediate comprehension of the robot intentions in a quick and intuitive way (e.g., making visible the robot's planned motion and task state) [20,21] or adapting the AR contents to the specific environmental or task conditions [22,23].

However, the majority of existing AR solutions looks at technology and robots, while neglecting the human aspects [10]. The main problem in AR-supported HRI is the lack of user friendly and intuitive interfaces implemented in accordance with the interaction design principles to guide users. While there are some attempts to design user-centered AR interfaces for different applications [24–28], for robotics applications, AR interfaces are usually developed by technology experts and not by UX designers. As a result, interfaces are technology-driven and not user-driven and they usually appear not fully centered on the users' perspective [10].

Only recently, a limited number of papers focused on the need to apply structured HCD methods to the design of HRI, focusing on the understanding and satisfaction of human needs. For instance, a UX-oriented methodology has been recently defined to investigate the human–robot dialogue and map the interaction with robots in performing shared tasks, eliciting the requirements for a valuable HRI design [10]. Similarly, another study has considered the role and relevance of UX in HRI and defined the actual trends concerning the inclusion of UX related to socially interactive robots [29]. Another work also proposes an innovative user-centered design tool to design AR platforms for maintenance operations [30]. Despite this, they did not specifically focus on the design of AR interfaces to support human–robot collaborative tasks, where a limited attention is paid to user perception, ergonomics, and usability issues. Contrarily, the nature of AR and the role that such applications can assume in the context of O4.0 requires great attention to human aspects. Interdisciplinary research is also advisable to achieve high-quality HRI.

In this context, the paper provides two main contributions:

1. A systematic review on AR-supported applications for human–robot collaborative tasks in industry, focusing on human aspects. As a result, the reader can understand whether and how UX approaches are currently adopted in the design of AR-supported collaborative solutions, as well as the main benefits and challenges of the application of UX methods in this field;

2. A UX-driven framework to design user-centric AR interfaces for industrial HRI, discussing also the main potential future developments, after having revealed the lack of such structured framework in literature.

## 2. Methodology

### 2.1. Systematic Literature Review

A systematic literature review (SLR) approach has been adopted to investigate the literature relevance of HCD and UX-based methodologies applied to HRI in collaborative tasks. Replicability and objectivity have been considered as basic principles in carrying out the research: the review follows the PICOC framework proposed by [18], thanks to its systematicity and completeness. PICOC [31] stands for a list of items to consider in the analysis, respectively *population, intervention, comparison, outcomes* and *context*. It has been chosen to outline the key concepts of the research. For this review, hereafter the considered items:

- *Population* consists of AR-supported industrial collaborative tasks;
- *Intervention* involves the HCD and UX approaches to design AR application for industrial collaborative tasks;
- *Comparison* can be done considering current design approaches and similar set-ups;
- *Outcomes* can be measured in terms of common Key Performance Indicators (KPI) like time to complete the operation, task's cognitive demand or physical workload;
- *Context* includes industrial human–robot applications.

## 2.2. Research Questions

The goal of the study is to provide a comprehensive overview of how UX has been used in the field of AR-supported collaborative applications for industry. Bearing this in mind, the authors formulated three research questions (Qi) according to the PICOC results:

- *Q1: What are the state of the art UX approaches in AR-supported collaborative solutions?*
- *Q2: What are the main benefits of adopting UX approaches in designing AR-supported collaborative solutions?*
- *Q3: What are the main challenges in designing AR-supported collaborative solutions?*

## 2.3. Search and Selection Process

The search was conducted on the Scopus database, since it encompasses different digital libraries, such as IEEE or ACM, and provides high quality, indexed papers. The inclusion criteria are:

- ○ Typology: the study considers articles on international journals and papers on conference proceedings, or books;
- ○ Topics: the study contains the keywords "*augmented reality*" + "*human robot interaction*" or "*human robot collaboration*" + "*user experience*" or "*user interface*". The search has been applied to "Title", "Abstract", and "Keywords" (TAK) fields. No reference to the "Mixed Reality" term was included since it subsumes both AR and VR;
- ○ Year: the study has not been limited in terms of the publication year.

According to [18], the following exclusion criteria have been defined:

- ○ Language: the paper is not written in English;
- ○ Scope: the paper is out of scope and focuses on different research domain;
- ○ Accessibility: the paper is not available.

Seeing the high specificity of the "*user experience*" and "*user interface*" keywords, the initial search returned 27 papers. No secondary documents nor patents were found. No further papers' selections in terms of field of application of AR-supported collaborative tasks were provided, since the main interest is analyzing the current general sensibility towards the UX approaches.

After the above-mentioned selection process, only 21 papers were admitted. The results from inclusion and exclusion criteria, and the number of papers found at each step, are shown in Table 1.

**Table 1.** Search and selection results.

| Search String | Database | Date | Found |
|---|---|---|---|
| TITLE-ABS-KEY ((augmented AND reality) AND (human AND robot AND interaction OR human AND robot AND collaboration) AND (user AND experience OR user AND interface)) | Scopus | 30/04/2021 | 27 |

| Exclusion Criteria | Found |
|---|---|
| Language | 27 |
| Scope | 23 |
| Accessibility | 21 |

Afterwards, the selected papers have been evaluated by a structured quality assessment procedure using three quality criteria (QC) similarly to [18]:

- QC1: It reflects the quality of the journal on which the paper is published, where Qi refers to the quartile score, according to Scimago Journal Ranking [32]. A score of 1 was assigned to Q1 journals, 0.5 to Q2, and 0.25 to Q3. If the journal belongs to Q4, or if it does not belong to a specific quartile yet or it is part of a conference proceeding, a "/" is assigned counting as 0;

- QC2: It reflects the relevance of the specific paper. A value of 1 is assigned if the paper specifically has "User Experience", "User Interface", or "Human-centered Design" as one or more paper keywords. This choice was made to further understand if the paper was intended to be searchable for UX, HCD, or UI-related topics;
- QC3: It reflects the citation impact. It considers the number of total citations of the paper (*c*) compared to the maximum number of citations of the most cited paper (*mc*) among those included in the review. Certainly, this criterion will not be quantitatively relevant for the most recent works, but it helps to understand the most significant works as recognized from the scientific community. As a consequence, a final score ranging from 0 to 1 has been determined for each paper (*i*) included in the review:

$$QC3(i) = c(i)/mc$$

The final aim is to focus the review attention on papers which match at their best the intentions of the authors and to allow the reader to select the best referenced articles according to such criteria.

## 3. Review Results

The results of the quality assessment are shown in Table 2, listed from the highest-quality paper to the lowest-quality paper: the final quality score is the direct sum of the results obtained according to the three considered criteria. The table highlights how there is still little attention given to the current topic, which has been tackled starting from the last few years. As for publications in the last year, it must be considered that still-ongoing studies have to be published and papers' impact in terms of citations need more time to be evaluated. The majority of collected studies are quiet below the half of the maximum allowed quality score, indicating that, according to quality criteria, there is still need of further research in this field. Considering QC2 scores, nearly half of the selected research papers do not include any reference to any of the HCD keywords, remarking the abovementioned lack of a UX sensibility.

As shown in Figure 2, in the last five years the attention towards AR solutions applied to robotics has noticeably increased. It can be related to the introduction on the market of several proprietary software development kits (e.g., ARKit, ARCore) from major vendors in 2017, which stimulated a general enthusiasm in AR applications development, and the contemporary commercialization of the Microsoft Hololens, from late 2016. These two facts pushed the academic and industrial interest on AR topics and potentials. In fact, previous tools did not provide effective augmented–cognitive interaction and lack in proactively supporting operators on receiving only the relevant information at their smart devices from nearby machines [33].

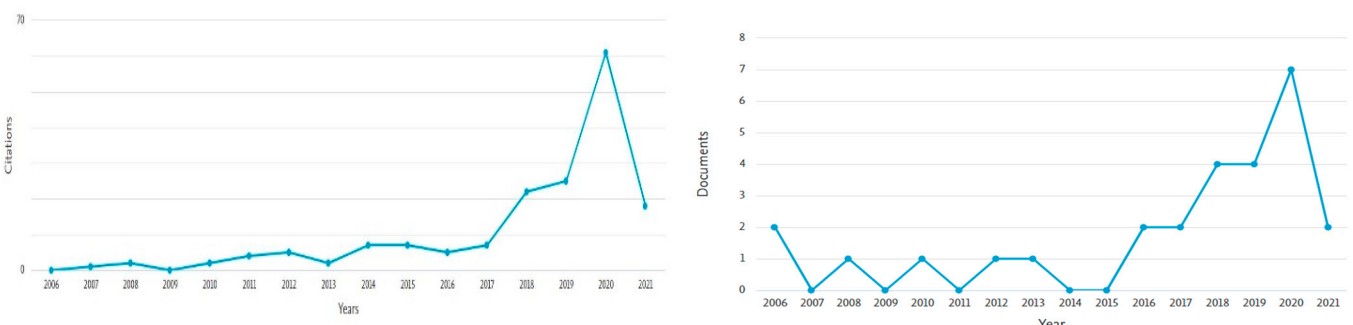

**Figure 2.** Papers by publication year (**on the left**) and total citations per year (**on the right**).

**Table 2.** Quality assessment on selected papers.

| Paper | Year of Publication | Publication Destination | QC1 | QC2 | QC3 | Quality |
|---|---|---|---|---|---|---|
| Hietanen, A., Pieters, R., Lanz, M., Latokartano, J., Kämäräinen, J.-K. [8] | 2020 | Journal | 1 | 1 | 0.53 | 2.53 |
| Papanastasiou, S., Kousi, N., Karagiannis, P., Gkournelos, C., Papavasileiou, A., Dimoulas, K., Baris, K., Koukas, S., Michalos, G., Makris, S. [34] | 2019 | Journal | 1 | 1 | 0.53 | 2.53 |
| De Pace, F., Manuri, F., Sanna, A., Fornaro, C. [18] | 2020 | Journal | 1 | 1 | 0 | 2 |
| Huy, D.Q., Vietcheslav, I., Gerald, S.G.L. [35] | 2017 | Int. Conference | / | 1 | 0.33 | 1.33 |
| Materna, Z., Kapinus, M., Beran, V., Smrž, P., Zemčík, P. [36] | 2018 | Int. Conference | / | 1 | 0.26 | 1.26 |
| Aschenbrenner, D., Li, M., Dukalski, R., Verlinden, J., Lukosch, S. [37] | 2018 | Int. Conference | / | 1 | 0.26 | 1.26 |
| de Tommaso, D., Calinon, S., Caldwell, D.G. [38] | 2012 | Journal | 1 | 0 | 0.13 | 1.13 |
| Bazzano, F., Gentilini, F., Lamberti, F., Sanna, A., Paravati, G., Gatteschi, V., Gaspardone, M. [39] | 2016 | Journal | / | 1 | 0.13 | 1.13 |
| Cao, Y., Wang, T., Qian, X., Rao, P.S., Wadhawan, M., Huo, K., Ramani, K. [40] | 2019 | Int. Conference | / | 1 | 0.1 | 1.1 |
| Materna, Z., Kapinus, M., Beran, V., Smrž, P., Giuliani, M., Mirnig, N., Stadler, S., Stollnberger, G., Tscheligi, M. [41] | 2017 | Int. Conference | / | 1 | 0.06 | 1.06 |
| Kyjanek, O., Al Bahar, B., Vasey, L., Wannemacher, B., Menges, A. [42] | 2019 | Int. Conference | / | 1 | 0.03 | 1.03 |
| Leutert, F., Herrmann, C., Schilling, K. [43] | 2013 | Int. Conference | / | 0 | 1 | 1 |
| Ji, Z., Liu, Q., Xu, W., Yao, B., Hu, Y., Feng, H., Zhou, Z. [44] | 2019 | Int. Conference | / | 1 | 0 | 1 |
| Frank, J.A., Moorhead, M., Kapila, V. [45] | 2016 | Int. Conference | / | 0 | 0.83 | 0.83 |
| Green, S.A., Chase, J.G., Chen, X.Q., Billinghurst, M. [46] | 2010 | Journal | / | 0 | 0.56 | 0.56 |
| Jones, B., Zhang, Y., Wong, P.N.Y., Rintel, S. [47] | 2020 | Int. Conference | / | 0 | 0.03 | 0.03 |
| Xin, M., Sharlin, E. [48] | 2006 | Int. Conference | / | 0 | 0.2 | 0.2 |
| Fuste, A., Reynolds, B., Hobin, J., Heun, V. [49] | 2020 | Int. Conference | / | 0 | 0 | 0 |
| Chan, W.P., Hanks, G., Sakr, M., Zuo, T., Machiel Van Der Loos, H.F., Croft, E. [50] | 2020 | Int. Conference | / | 0 | 0 | 0 |
| Krauß, M., Leutert, F., Scholz, M.R., Fritscher, M., Heß, R., Lilge, C., Schilling, K. [6] | 2021 | Journal | / | 0 | 0 | 0 |
| Diehl, M., Plopski, A., Kato, H., Ramirez-Amaro, K. [51] | 2020 | Int. Conference | / | 0 | 0 | 0 |

Figure 3 depicts the main subject areas dealt by the selected papers. One can infer that the design of AR applications supporting collaborative tasks do not merely involve engineering considerations on technologies or infrastructure's deployment, but also other field of study, such as psychology, neurosciences, and social sciences (i.e., tackling interaction issues from the human point of view or determining the most useful physiological parameters to consider in the evaluation of a specific interface).

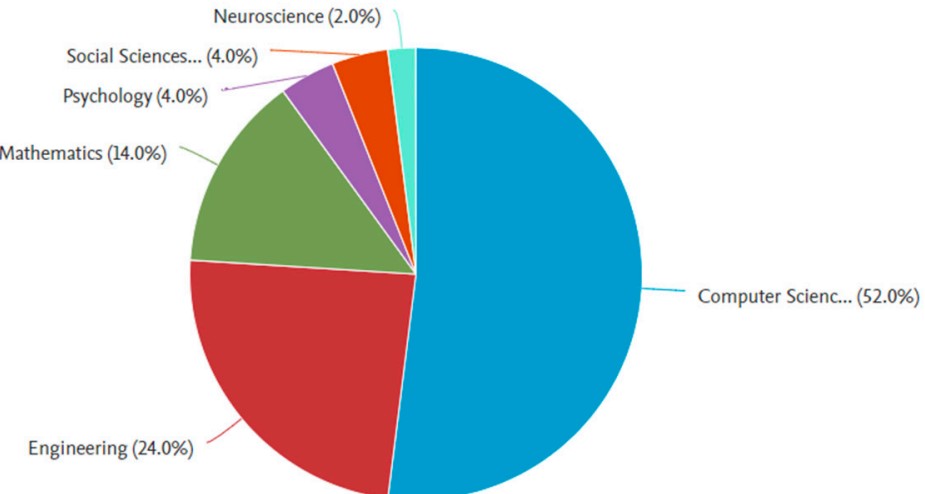

**Figure 3.** Papers by subject's area.

In conclusion, the relevant works being selected have been carefully considered against the research questions (Q1, Q2, and Q3) as presented in the following sections.

### 3.1. What Are the State of the Art UX Approaches in AR-Supported Collaborative Solutions?

The current trends in the design of collaborative tasks supported by AR technologies do not systematically show a great attention to UX topics. Hietanen et al. [8] proposed an interactive user interface to assist O4.0 in performing robot-assisted tasks comparing two separate implementations of the same system: a projection-mirror setup, and a wearable device (i.e., Microsoft Hololens). No prior UX assessment was proposed; as part of the subjective evaluation, a final questionnaire including 13 questions divided into six categories (respectively: safety, information processing, ergonomics, autonomy, competence, and relatedness) was submitted to roughly understand mental and physical stress. Comments from users were collected to deepen the subjective impression, without using any structured method to collect the perceived workload, as used for instance in different contexts. A more structured approach is presented by Papanastasiou et al. [34]: the paper emphasizes the need of a seamless integration between the human operator and his robotic counterpart by monitoring both working entities through sensors and wearable devices. This led to the re-design of the workplace from the human point of view to promote both the robot's operability and operator's mobility, without any barrier to separate them; a multi-stage iterative process has been followed, starting from technical and functional specifications as well as safety requirements. A digital simulation is included for supporting cell setup and risk assessment.

De Pace et al. [18] placed attention on AR devices' usability as enabling tools. The authors reported how usability, workload, and likability can be investigated thanks to the standardized questionnaire (e.g., NASA-TLX [52], System Usability Scale (SUS) [53], AttrakDiff [54]). The same intention is expressed by Huy et al. [35], who introduced a novel AR handheld device inspired by the abovementioned multimodality perceptive interface, incorporating hand gesture mapping, haptic buttons, and laser pointers. The system can suggest available options to the operator and wait for a response instead of traditional inputs by keyboard or mouse; a usability investigation is eventually foreseen to improve the interface effectiveness with the help of user's feedbacks.

Materna et al. [36] evaluated the idea of spatial augmented reality (SAR) through a UX study. The outlined approach works towards avoiding continuous switching of attention during demanding tasks, thanks to a correct distribution of information along the operations and a shared workplace, to be usable also for non-expert users. Process simplification was also addressed by Aschenbrenner et al. [37] to reduce the installation time of hybrid robot–human production lines, and by De Tommaso et al. [38] that defined a

new process of skill transfer between human workers and robots. Similarly, Fuste et al. [49] presented a holistic UX framework (called "Kinetic AR") for visual programming of robotic motions using AR: the goal was to guarantee a low entry barrier to intricate spatial hardware programming. The UX approach was achieved through interviews to robotic system integrators, manufacturers, and end-users with different expertise, to finally identify the goals and requirements to be accomplished. Communications and interactions were also investigated by Bazzano et al. [39], using 3D immersive simulation to support the design and validation of natural HRI in generic usage contexts, comparing an AR interface and a non-AR one. Among others, subjective observations were gathered through the SASSI methodology [55] to evaluate speech interaction in both interfaces. Information on completion times, overall satisfaction, ease of use, perceived time requested, and support information were collected, and their statistical relevance was given by running an independent sample *t*-test.

As a result of the review, one can state that there are few preliminary attempts to include UX in the design of AR applications for HRI purposes, as summarized in this paragraph, but a reference, ready to use model that is able to integrate the users' subjective evaluation and the analysis of the quantitative human–robot performance is still missing.

The main weaknesses of the current attempts are:

- User testing is usually based on the collection of deconstructed data regarding device or interface usability, system likability, cognitive and physical workload, or the overall subjective sense of safety in performing the selected operation, without a robust reference model;
- Even if a good attention in using multimodal interfaces to optimize HRI is arising, this trend is not mature enough to enhance human sensorial capabilities by integrating different sensors (e.g., force/torque sensors, microphones, cameras, smartwatches, and AR glasses);
- AR application design does not consider the user perspective and does not help in the improvement of the ease of use of industrial workplaces, avoiding uncomfortable conditions (e.g., extra lightning and noise).

These results highlight the need of a structured framework to design AR interfaces for HRI and pushes towards its definition.

### 3.2. What Are the Main Benefits of Adopting UX Approaches in Designing AR-Supported Collaborative Solutions?

After the first analysis, the review focused on the analysis of the benefits related to the adoption of UX-based approaches in the design of AR applications for HRI: these approaches generally turn into a detailed evaluative UX phase, where subjective questionnaires represent the main source of information. Table 3 summarizes the most significant papers dealing with such an aspect, also reporting the main areas of applications.

Within the context of laboratory object manipulation tests, Frank et al. [45] focused on the user interaction effectiveness of a mobile augmented interface and on virtual graphics appearing as task's visual cues to reduce cognitive burden on end-users. The proposed system can automatically intercept an operator's intention on virtual objects (i.e., drag and drop of models in the space), thus reducing the human involvement while operating with the collaborative companion. No defined UX approach was adopted: a revision of the overall interface was conducted through a final questionnaire after the user-testing phase to identify possible criticalities. A concurrent interface simplification without losing its functionalities in the human–robot collaboration is indeed of extreme importance, in opposition to what has been defined by [42], where high cognitive functionalities are purposely omitted from the proposed interface.

A further critical point in AR-supported collaborative tasks is the choice of the correct interface to use, which is usually conducted without a precise validation tool or methodology. In De Pace et al. [18], a series of interesting UI studies resembling HCD approaches were collected concerning whether exocentric or egocentric interfaces are the best in limit-

ing the level of mental and physical involvement in controlling the manipulator. Another study by Chan et al. [50] reconsidered AR-based interfaces for human–robot collaboration on large-scale labor-intensive manufacturing tasks (carbon-fiber-reinforced-polymer production) where the accent is on the perceived workload and efficiency. Indeed, as stated in other studies [18], the lowest physical and temporal demand is registered with appropriately designed AR solutions, reducing user's effort and sense of frustration while cutting down operational time. Such an approach does not explicitly make reference to a structured and systematic HCD methodology, but it relies on NASA-TLX questionnaire results. Similar conclusions were reported by Diehl et al. [51], where application circumstances for the choice of best device are examined, starting from users' feedback on robot's time and area of manipulation up to user sense of safety.

In Xin et al. [48], a collaborative task concerning playing board games was explored and evaluated by examining various interaction opportunities arising when humans and robots collaborate. This interesting analysis was related to two contrasting robotic behavioral conditions which have been tested: a human-centric condition where robot behavior is more accustomed to human obedience, and a robot-centric one where suggestions coming from the operator are neglected. Statistical results on the final user testing phase related to a custom questionnaire allows for a reinforced idea of the centrality of a human-centric condition to increase the sense of collaboration of O4.0.

Moreover, Palmarini et al. [56] stressed that safety is deemed as one of the most relevant aspects in human–robot collaborative systems and context-awareness information is unavoidably important to enhance user perception. Analogously, Quintero et al. [57] proposed two separate approaches to draw AR paths, respectively, a free space and a surface trajectory one. Such proposals could be effectively integrated to optimize robot's programming phases with a UX sensibility, reducing programming time, and allowing the worker to selectively inspect different robot trajectories and to work on them in a user-friendly interface. For an optimal collaboration, robot intention is another source of essential information within a HCD approach: a general indifference on the topic emerges from actual selection, although Liu et al. [58] described a temporal and-or graph (T-AOG) to allow the human understanding of the robot's internal decision-making process, to supervise its action planner, or to monitor its latent states (i.e., forces and moments exerted while interacting).

**Table 3.** Papers focusing on added value related to adoption of UX approaches in HRI.

| Paper | Benefits | Adopted UX Tools | Area of Application |
|---|---|---|---|
| J. A. Frank, M. Moorhead, and V. Kapila [45] | End-user's intentions understanding to reduce operator cognitive burden | Custom questionnaire | Object manipulation |
| W. P. Chan, G. Hanks, M. Sakr, T. Zuo, H. F. Machiel Van Der Loos, and E. Croft [50] | The system's final application must be considered to prevent wrong choices in terms of interfaces and to avoid physical and cognitive repercussion on the user | NASA-TLX | Large-scale, labor-intensive manufacturing tasks |
| C. P. Quintero, S. Li, M. K. Pan, W. P. Chan, H. F. Machiel Van Der Loos, and E. Croft [57] | Reducing robots' programming operation time and cognitive demand | Custom questionnaire | Robot programming |

As emerged from the review findings, several benefits derived from a UX-based approach when implementing AR-supported collaborative tasks, both objective and subjective: a systematic cognitive and physical relief on the operator, an increased working efficiency, a reduction in operational time and sense of frustration when interacting with shop floor interfaces, and an improved sense of safety and inclusiveness while collaborating with the robotic counterpart. Such conclusions were mainly reported after a user testing campaign in which standard or customized questionnaires were designated to collect final tester impressions to be subsequently reanalyzed by the papers' authors.

*3.3. What Are the Main Challenges in Designing AR-Supported Collaborative Solutions?*

The literary review highlighted that the design, development, and use of AR technologies to improve HRI in industrial contexts is a hot topic from a technological point of view; however, there is a lack of models to deepen the UX and only a limited number of papers have proposed the adoption of UX methods to support the design of AR application in this field, according to user-centered principles. As reported by recent market forecasts, the mixed reality market size (including both augmented and virtual reality technologies) is expected to grow by USD 125.19 billion during 2020–2024 [59], up to USD 1.45 trillion to by 2030 [60]. This rapid growth entails big challenges from both a technical and technologies viewpoint and a human viewpoint. Some issues, just considered so far, need to be investigated and faced: from privacy problems to safety requirements. This means that the design of AR-supported applications in the context of HRI will consider how to manage the robots' and operators' data collected and how to assure the proper privacy and safety levels. Considering current applications [61], one can reflect on both critical success factors and challenges related to future robust industrialization. If compared to industrial software systems, current AR hardware readiness seems to still be far from a mature adoption in industry. Thus, human-centered design methods are required to balance industrial system requirements with human needs and social concerns; in this sense AR is so close to human abilities, also affecting and empowering them. Another challenge is the integration of AR devices within modern manufacturing systems: data exchange to and from the AR application should be compliant with robotics and automation standards to assure a full adoption in industry. In regard to this topic, only few research attempts have been made (e.g., AutomationML [62]) which are still far from the inclusion of AR data.

Moreover, a proper UX evaluation framework for AR-supported collaborative tasks needs to be defined. A first attempt has been made considering UX analysis in the design of HRI applications using a structured approach [10], but not including AR tools. On this topic, the main challenge is to define a systematic and coherent way to interpret data coming from different equipment and returning AR digital contents to the O4.0, in an adaptive and intelligent way, considering the UX, and further enhancing the human physical, sensorial, and cognitive capabilities by means of human cyber–physical system integration [63]. In this direction, a further challenge is promoting socially embedded human–robot collaborations where human communications can be used to adapt service robots to the user needs accordingly: it consists of giving the robots the concept of emotional tuning and to emphatically communicate with machines [64].

Moreover, the estimation of those variables affecting trust in HRI is necessary to design new, effective AR interfaces providing situational awareness and spatial dialog, and to determine functional elements to improve human confidence in robots. This evaluation should be included in a comprehensive approach considering validated metrics for an overall UX assessment [65]. Contextually, the assessment of human cognitive and physical efforts in developing collaborative tasks has an absolute relevance.

Finally, in the context of AR-supported human–robot collaborative operations, user testing needs a more statistically reliable base, including both academic and industrial studies and increasing the number and typology of people involved to assess the effectiveness of AR in HRI tasks [18]. The results mainly imply the definition of new UX-based methods to design AR interfaces from a multiple users' point of view, involving novice and expert users, and the benchmark of the most suitable wearable interfaces to be used together with industrial robots.

## 4. Discussion on Review Results

From the current analysis, a substantial lack of structured methods to design user-centered AR applications for HRI have emerged. Despite several attempts in other various contexts (tourism, mobile application games, etc. [24–28]), UX-driven methodology are poorly adopted in HRI, and this led to the design of interfaces which are far from real users' needs.

On this base, the authors believe that it is crucial to promote UX-driven design processes to develop successful AR interfaces, especially for collaborative tasks involving humans and robots, to fully support the O4.0. Only the adoption of a proper HCD framework allows considering the real needs of the operators in a specific context of use, focusing not only on safety and ergonomics issues, but on the overall UX aspects (e.g., usability, intuitiveness, satisfaction, cognitive load, and emotional response) with the final aim to have a renovated human–robot relationship where both subjects are actively participating and transmitting knowledge to the equivalent counterpart, according to a win–win approach.

For these purposes, the review results suggest defining a structured UX-driven process to design AR interfaces for HRI tasks. The review represents a nonlinear and iterative process aiming at assisting the AR interface designer in implementing a valuable communication with robots, during the execution of HRI tasks. As depicted in Figure 4, the process must be made up of three main steps:

1. Requirements Gathering;
2. AR Interface Design and Prototyping;
3. UX Assessment.

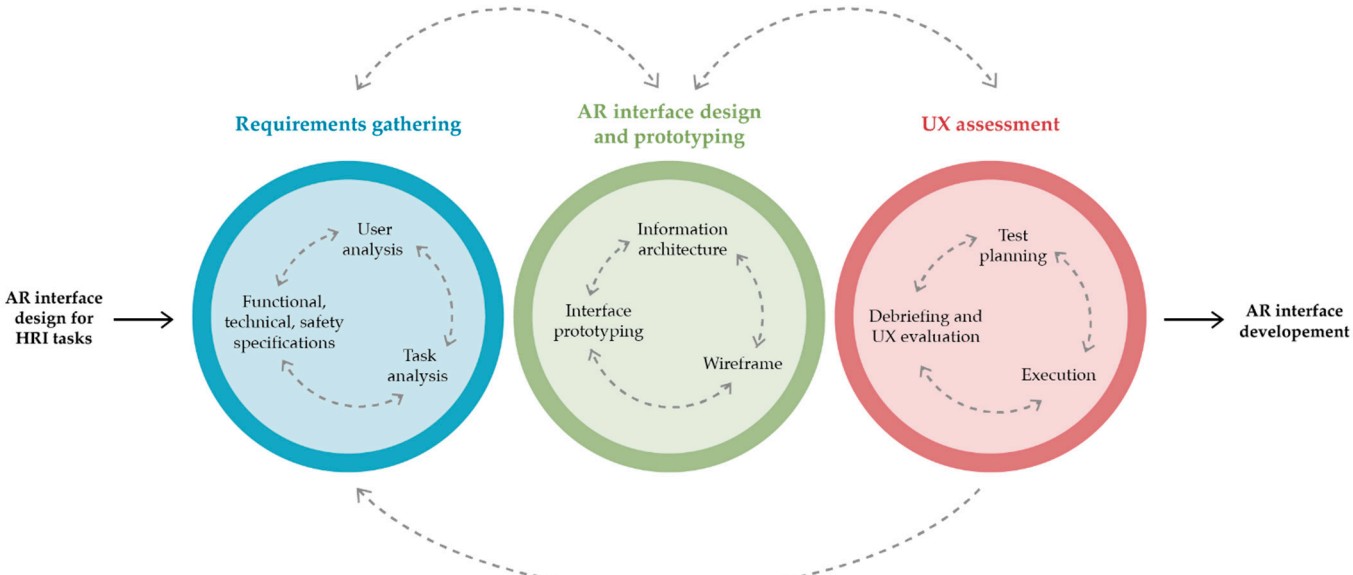

**Figure 4.** The need of a UX-driven process for AR interface design to support HRI tasks.

The process starts with the need to define a new AR interface for collaborative tasks, and brings to the interface definition, before the AR interface development. Such a process goes a step further with respect to a few recent studies [10,34].

The first step to be investigated consists of requirement gathering from user research. It is a vital part of any UX-driven processes because it is the act of understanding the users and their needs, making sense of who the user is, what he/she wants, and how he/she will perform a certain task. It mainly consists of a deep, accurate user analysis based on the context research to be carried out in a not invasive way using a set of UX design techniques. The most suitable techniques for user understanding are user observation, focus groups, and interviews [66]. User observation consists of observing users in their natural environment without affecting their normal behaviors and performance, with the aim to understand the users' needs and widely used when users belong to specific categories. It can also be done using video analysis, but in some industrial contexts video recording is not always possible. Such analysis is frequently combined with focus groups or interviews, which can provide a deeper insight about the users' habits and behaviors since they actively involve users to collect, in different ways, qualitative data about their

needs, expectations, or fears. The mixed approach combining different techniques is very useful since it allows combining quantitative and qualitative data. After that, the list of executed tasks can be easily defined, reporting also the actors involved for each task (e.g., humans, robots) and the time span. These actions finally led to the definition of design specifications considering functional, technical, and safety aspects. A different set of UX design techniques can be validly adopted, from user scenarios and personas to experience maps, as proposed by [10]. User scenarios are stories to show how users might act to achieve a goal in a system or environment. They are valuable aids for designers to visualize aspects of their solutions which users might appreciate most in their contexts of use and with their unique needs and motivations. Personas represent the target users by a set of probable users and flesh out their experiences to reflect realistic situations. Finally, experience maps represent a synthetic visualization of an entire end-to-end experience that generic users (i.e., personas) go through to accomplish a certain goal, and they allow a better understanding of human behaviors.

The second aspect to be investigated is the AR interface design and prototyping. Design consists of the definition of the interface functions as well as the items, while prototyping presents the design in a concrete way by representing the interface in action with the simulation of the final interaction between the user and the system. In this context, wireframes are very powerful as a visual representation of the interface pages; clickable wireframes are the simplest form of interactive prototype, created by linking static wireframes together. Moreover, using digital tools, wireframes can be updated and easily reused and layouts can be easily changed based on user feedback to repeat the testing process. Low-fidelity prototypes allow one to easily define the following information architecture, also comparing possible alternative solutions, and to optimize the design itself in an iterative way.

Finally, the third aspect to include consists of the UX assessment. Different evaluation techniques exist to investigate more closely the user's behavior and perception on a final prototype or products. One of the most spread is user testing, that can lead to both quantitative and qualitative data [67]. Quantitative tests carry out measurements (e.g., execution time, number of errors, and number of tasks completed) while performing a specific task on the user interface. User testing sessions often include post-test evaluation questionnaires (e.g., System Usability Scale (SUS) [53], UEQ Questionnaire [68], or meCUE questionnaire [69]) and allow the gathering of many opinions in a short time, as well as being adaptable to multiple application areas. In addition, physiological measurements allow us to investigate in real time the level of physical and cognitive workload as well as stress of the operator during the interaction. Examples of adoption of these tools for the design of modern systems is provided by [70].

Based on the review results, it is also possible to identify some trends of future implementation of a successful UX-driven design process, as depicted above. First of all, the UX assessment needs to be included in the analysis of the real human activity during collaborative task execution, to understand the level of attention and physical and cognitive responsiveness through wearable devices, in order to understand the real UX. Secondly, non-intrusive sensors and smart interfaces are required to carry out a bi-directional communication flow from-to machines and robots and create a synergic collaboration, in order to overcome the current vision based on separate entities with incompatible characteristics. Further, flexible data management is necessary to manage the crescent complexity and the larger amount of data coming from the shop floor and the operator, in order to successfully integrate AR-supported collaborative tasks in the overall productive chain. Finally, it is worth understanding the impact of AR on user perceptions, ergonomics, and human–robot interactions.

## 5. Conclusions

This paper reviews the overall condition of industrial collaborative tasks supported by AR technologies, pushed by the growing interest of industry in the AR market registered

in the last 5 years and the need to define new ways to make the Operator 4.0 successfully work in the factories of the future, interacting with robots. In this context, AR interfaces can help to improve human–robot communication and interaction at different levels, thanks to the possibility to show contextual and digital information and data when and where needed. However, there is a lack of a proper framework to design AR interface, including the operators' UX, specifically designed for HRI. The paper starts with a review of the state of the art, focusing on the inclusion of HF and UX design principles in the design of AR interfaces to support HRI. In the paper, a SLR approach was used to collect the most interesting papers in this field, considering the recent scientific literature. After identifying the focus of the current study, 27 papers were gathered and assessed according to proper quality control parameters previously defined. At the end of the selection process, 21 papers were deemed suitable to answer the three research questions: *Q1: What is the state of art related to UX approaches in AR-supported collaborative solutions?; Q2: What are the main benefits of adopting UX approaches in designing AR-supported collaborative solutions?; Q3: What are the main challenges in designing AR-supported collaborative solutions?*

As a result, the research highlighted the lack of reliable and systematic user-centered methodologies to design AR applications for human–robot collaborative tasks. This fact is limiting the acceptance of such solutions and slowing down the technological integration of smart devices within the Operator 4.0 paradigm. Several added values of AR application to Operator 4.0 scenario are then presented, starting from the reduction of the worker 's cognitive workload thanks to the interface simplification and adequate usability tests, up to the realization of a shared workplace where a synergic collaboration could take place, in which both actors can reciprocally be understood and learn from the corresponding counterpart. From the discussion of the review results, the paper finally highlights the need of structured UX-driven processes to design successful AR interfaces for human–robot collaborative tasks, made up of different phases organized in an iterative cycle, including typical UX design tools and techniques for interface design, that are not currently used in AR interface design for industrial purposes. The research also defined the main trends of development for future applications, considering the need of non-intrusive human monitoring devices and smart tools to enable fruitful communication between operators and the on-going process at the shop floor.

**Author Contributions:** Conceptualization, M.P. (Margherita Peruzzini) and R.K.K.; methodology, M.P. (Margherita Peruzzini) and E.P.; formal analysis, R.R.; data curation, R.K.K.; writing—original draft preparation, R.K.K. and M.P. (Margherita Peruzzini); writing—review and editing, M.P. (Marcello Pellicciari) and R.R.; supervision, M.P. (Marcello Pellicciari). All authors have read and agreed to the published version of the manuscript.

**Funding:** This research received no external funding.

**Institutional Review Board Statement:** Not applicable.

**Informed Consent Statement:** Not applicable.

**Data Availability Statement:** All available data is contained within the paper.

**Conflicts of Interest:** The authors declare no conflict of interest.

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
