# Peer review of "UX in AR-Supported Industrial Human–Robot Collaborative Tasks: A Systematic Review"

_applsci, doi:10.3390/app112110448_

Round 1

Reviewer 1 Report

In this resubmitted manuscript, the previous comments were addressed, and I cannot find any problem with the manuscript.

Author Response

Thanks!

Reviewer 2 Report

Accept

Author Response

Thanks!

Reviewer 3 Report

The paper presents a literature review on user experience design in the design of AR interfaces for human-robot interface. The topic is current and very relevant for the development of assistive technologies required in the Industry 4.0 context.

In addition to the systematic review, the manuscript proposed framework to design AR interfaces. The framework seems a first step towards a structured method to design AR applications for HRI systems, but its contribution could be significantly improved if the framework presented a standardized/recommended set of tools to be used in each step of the process.

The topic of the paper is clear. The manuscript is logically structured in a clear narrative. Title is appropriate and matches the content of the paper. Even though some corrections need to be addressed throughout the paper, it reads well and the ideas are clearly transmitted to the reader. References are recent, relevant and appropriate in number.

Please, consider reviewing the following:

  • In general, it is best to avoid using abbreviations and acronyms in the abstract (O4.0, line 10).
  • The acronym O4.0 is used in lines 49 and 51, but it is only defined in line 55.

General remarks

  • SCOPUS is indeed a relevant database with high quality papers. But why other relevant databases, such as Web of Science and Dimensions, were not included in the search? The number of exclusive content in each database may be low, but considering that the systematic review returned in only 21 papers the addition of other sources may not add significant effort.
  • Reference [15] does not defined exclusion criteria (lines 229-233) and does not perform a structured quality assessment procedure (lines 247-248). Are you referring reference [18] instead?

Author Response

We would like to thank the reviewer for the precious comment provided. We have carefully revised the entire manuscript and removed abbreviations from the abstract: Operator 4.0 definition was anticipated as You suggested for a clearer reading and manuscript understanding.

As for the choice of the reference database, Scopus was considered since research on other databases like Dimension and Web of Science returned limited results which were not different  from the Scopus one: Scopus indeed offers more tools for a detailed and systematic review.

Reference [15] was a typo and as You correctly understood it is referred to [18] instead.